# Assessment of the Effective Tissue Concentrations of Injectable Lidocaine and a Lidocaine-Impregnated Latex Band for Castration in Calves

**DOI:** 10.3390/ani14060977

**Published:** 2024-03-21

**Authors:** Joseph A. Ross, Steven M. Roche, Kendall Beaugrand, Crystal Schatz, Ann Hammad, Brenda J. Ralston, Andrea M. Hanson, Nicholas Allan, Merle Olson

**Affiliations:** 1Chinook Contract Research Inc., Airdrie, AB T4A 0C3, Canada; joe.ross@ccr01.com (J.A.R.); kendall@ccr01.com (K.B.); crystal.schatz@ccr01.com (C.S.); ann.hammad@ccr01.com (A.H.); nick.allan@ccr01.com (N.A.); 2ACER Consulting Ltd., Guelph, ON N1G 5L3, Canada; sroche@acerconsult.ca; 3Applied Research Team, Lakeland College, Vermillion, AB T9X 1K5, Canada; brenda.ralston@lakelandcollege.ca (B.J.R.); andrea.hanson@lakelandcollege.ca (A.M.H.); 4Alberta Veterinary Laboratories Ltd., Calgary, AB T2C 5N6, Canada

**Keywords:** EC_50_, EC_95_, anesthetic, elastrator, pain control, ring block, pharmacokinetics, pharmacodynamics

## Abstract

**Simple Summary:**

Castration is commonly performed in young dairy calves and results in pain and discomfort. This study aimed to assess the effective tissue concentrations of the current standard of care for pain mitigation in calves during castration (injectable lidocaine) and to assess the ability of lidocaine-loaded bands (LLBs) to deliver effective concentrations into the scrotal tissue over time. The injectable lidocaine provided effective anesthesia for up to 60 min, highlighting the importance of finding additional strategies to manage long-term pain. A ligation band impregnated with lidocaine could provide a suitable alternative, as it appears to deliver effective lidocaine concentrations starting as early as 2 h following application and lasting at least 28 days after application. Further studies are warranted to compare the use of LLBs to injectable local anesthetics.

**Abstract:**

This study aimed to assess the effective tissue concentrations of the current standard of care for pain mitigation in calves during castration (injectable lidocaine) and to assess the ability of a lidocaine-loaded elastration band (LLB) to deliver effective concentrations into the scrotal tissue over time. This study comprised two different trials: (1) effective concentrations of injectable lidocaine in the scrotal tissue; and (2) the in vivo delivery of effective concentrations of lidocaine from LLBs placed on the calf scrotums. Sensation in the scrotal tissue was assessed by electrocutaneous stimulation. Injectable lidocaine allowed for short-term anesthesia for up to 60 min, highlighting the importance of finding additional strategies to mitigate long-term pain. An elastomeric ligation band impregnated with lidocaine could provide a suitable alternative, as it yielded tissue levels of lidocaine that approached EC_50_ and exceeded EC_95_ at 2 and 72 h following application, respectively, and remained above those levels for at least 28 days after application. Further studies are warranted to compare the use of LLBs to injectable local anesthetics.

## 1. Introduction

Castration is a non-health-related procedure performed on male calves to reduce aggression, unwanted breeding, and improve meat quality. Beef calves are typically castrated within days of birth until approximately 6 months of age [1]. Recently, there has been a large shift in the use of beef semen in the dairy industry, resulting in more surplus dairy calves entering beef production where castration is required [2]. Given this substantial increase in the number of calves requiring castration, there is a need to understand how this procedure affects their welfare.

It has been established that all castration procedures result in pain [3]. Specifically, no matter the method used to castrate, neurohormonal and electroencephalographic stress responses have been found [4]. Furthermore, it was found that older calves have a more pronounced stress response, highlighting the need to complete this procedure at an earlier age [4]. Due to the pain associated with this procedure, pain control has become a requirement in several quality assurance programs and Codes of Practice worldwide. For example, the Canadian Code of Practice for the Care and Handling of Veal Cattle, which encompasses the rearing of dairy–beef calves, states for castration that “At any age, pain control must be provided in consultation with a veterinarian, including local anesthesia and systemic analgesia” [5]. Multimodal pain management, using a local anesthetic and non-steroidal anti-inflammatory drugs (NSAIDs), has been shown to alleviate pain associated with the procedure in the hours and days following the procedure [6,7]. However, the efficacy of pain control is highly variable depending on the method used. Moreover, the repeated re-administration of NSAIDs would be required to maintain their efficacy over a longer period (e.g., days to weeks), which is likely not practical.

The application of rubber bands, which interrupt the flow of blood to the scrotum and testes, causing the necrosis of the tissue, has become a common method for castration in calves as it is bloodless, inexpensive, effective, and easy to apply to young calves [8]. Although this method initially leads to fewer obvious signs of pain [9], the pain related to band castration is chronic as the wound-healing process can last for weeks [10,11]. Furthermore, when compared to the use of surgical castration, dairy calves castrated with a rubber ring gained less weight and spent less time lying down over the 8-week study period [12]. This highlights the need for better methods to control the long-term pain associated with this castration method.

Effective concentrations of local anesthetics, such as the tissue concentration yielding a 50% or 95% reduction in sensation (EC_50_ and EC_95_, respectively), are important measures of potency [13,14,15]. Additionally, the time of onset and the duration of local anesthesia can be established by measuring the tissue concentration over time and comparing it to the EC_50_ [16,17]. Lidocaine is a well-studied local anesthetic in humans [18,19,20]. However, despite a small number of reviews on the effects of lidocaine in calves [1,21], there is little evidence regarding its pharmacokinetics, pharmacodynamics, and effective concentrations in the scrotal calf tissue. It is widely recommended—and often required (see, for instance, the Canadian Codes of Practice for the Care and Handling of different livestock species [22])—that pain control (such as local anesthetics) be used for painful procedures. More research is therefore needed to provide a more comprehensive understanding of the use of lidocaine as a local anesthetic in livestock species.

Chinook Contract Research Inc. has developed a latex elastration device in which lidocaine has been directly impregnated (patent numbers US11596510B2, CA3072762C, AU2018313951B2, NZ2762352A, EP3664720A4, and WO2019032928A1). These bands have been formulated to contain therapeutic levels of lidocaine, a local anesthetic, to allow for sustained local pain relief to be delivered to the castration area [23,24]. Preliminary work found that these lidocaine-loaded bands (LLBs) were able to deliver therapeutic levels of lidocaine into scrotal tissues over 7 days; however, additional validation work is required.

Hence, the hypothesis of this study was that electrocutaneous stimulation could be used to assess the effective tissue concentrations of a current standard of care for pain control during castration in calves (injectable lidocaine) and that LLBs would deliver concentrations of lidocaine into the contacted scrotal tissue that would meet or exceed these levels over time.

## 2. Materials and Methods

These studies were conducted in compliance with the animal care guidelines established by the Canadian Council of Animal Care and were approved by the Chinook Contract Research IACUC (A8217-03). Two different trials were conducted: (1) effective tissue concentrations of injectable lidocaine in the scrotal tissue; and (2) the in vivo delivery of lidocaine from LLBs placed on calf scrotums. The latex rubber band used as the LLB contained 80 mg of a lidocaine base USP (Lidoband^TM^, Solvet, Calgary, AB, Canada).

### 2.1. Effective Concentrations (EC_50_ and EC_95_) of Injectable Lidocaine in Calf Scrotal Tissue

Six intact male calves that were ~1 month of age, obtained from commercial sources, were enrolled in a castration trial after a period of acclimatization of at least 1 week following arrival. The calves were placed into individual pens (2.1 m × 0.76 m) with a partially slatted floor for waste removal. All animals were housed in the same room of a barn with forced air ventilation. The calves were vaccinated with an 8-way clostridial vaccine (Tasvax 8, Merck Animal Health, Kirkland, QC, Canada), respiratory disease vaccine (Inforce 3, Intranasal vaccine Zoetis Canada Inc., Kirkland, QC, Canada), and topical parasiticide (Solmectin Pour-On for Cattle, Alberta Veterinary Laboratories Ltd., Calgary, AB, Canada). The calves were fed twice daily (morning and afternoon) via 7.6 L buckets attached to the front gate of each calf’s individual pen. One bucket contained milk replacer, 5 L per day (Mapleview Functional Protein Blend + Deccox, Mapleview Agri Ltd., Drayton, ON, Canada), and the second bucket a dry pelleted complete feed (21% Protein Calf Starter, Landmark Feeds, Strathmore, AB, Canada). Water was freely available via nipples. At enrollment, the calves were required to be healthy based on a physical examination and could not have been previously treated with lidocaine, similar analgesics, or NSAID compounds. Prior to castration, six injection sites were marked at the scrotum neck of each animal with a 1 cm circle using a permanent marker (Figure 1).

At each site on the scrotal neck tissue, the calves received a 1 mL subcutaneous injection of 2% lidocaine-HCl lacking epinephrine (Teligent Canada Inc., Mississauga, ON, Canada) percutaneously, forming a “ring block” [25]. The location of the sites where the calves were injected was randomized for each animal and marked with colored markers. After injection, samples were collected from the center of the injection site area (defined by the colored marking) at T = 30, 60, 90, 120, 180, and 240 min using a punch biopsy (Acu-Punch, 4 mm diameter, 7 mm depth; Acuderm Inc., Ft. Lauderdale, FL, USA). Each biopsy from the sampled area contained both skin and subcutaneous tissue. At each time point, samples were collected from one of the six injection sites from all 6 animals in accordance with the randomized sampling schedule. To provide a lidocaine-free control tissue, two additional animals underwent surgical castration, with biopsy samples taken from their scrotums thereafter. All biopsies were frozen until being analyzed for the lidocaine content, as per Section 2.3 below. Following study completion, all calves were surgically castrated, treated with oral meloxicam (1 mg/kg body weight; Solvet, Calgary, AB, Canada), and observed for 3 days for adverse events (none were observed).

To determine the effectiveness of the local anesthetic activity, the response to electrocutaneous stimulation was graded according to the calf’s positive avoidance response, as previously demonstrated [23,24,26,27]. Specifically, infant monitoring electrodes (Red Dot 2258, 3M Canada, London, ON, Canada) were applied to the injection site, defined by the colored 1 cm permanent marker circle, and stimulated using a peripheral variable output nerve stimulator (Sun Stim Peripheral Nerve Stimulator, SunMed, Largo, FL, USA). Stimulation was completed immediately prior to the biopsies in the same colored circle. The peripheral nerve stimulator output was adjustable from 0 to 250 mA (frequency of 100 Hz). For the baseline and post-treatment readings, the target tissue was stimulated at five increasing stimulus levels (median current; range (mAmp)): 2 (<30), 4 (90; 87–95), 6 (160; 155–162), 8 (219; 211–225), and 10 (244; 243–245), until either the maximum level was reached or a positive avoidance response was observed, according to Table 1. Note that the response scores at each time post-injection were not normalized to the baseline scores. Note also that the assessment of the positive avoidance response was reached by the consensus of three individuals who were not involved in the initial injection and were unaware of the contents of the syringe.

### 2.2. In Vivo Delivery of Lidocaine into Scrotal Tissues from LLBs

To evaluate the in vivo release of lidocaine from the LLBs, 25 intact male calves weighing <200 lbs at approximately 1 month of age were enrolled into the study at a commercial calf-raising facility on 1 September 2021. Prior to enrolment, all calves underwent a physical exam and were enrolled if they were deemed healthy, were not previously treated with lidocaine or NSAIDs, and had not exhibited pre-study-complicating disease conditions that could interfere with or prevent evaluations and analyses in the study. Cryptorchid calves and calves with inguinal hernias were not included in the study. Field technicians were blinded to the identity of the control bands versus the LLBs. At time 0 (T = 0), an LLB was placed around the scrotal neck of 20 calves, and a control band (i.e., no lidocaine) around the scrotal neck of 5 calves, using a banding tool. Note that the animals were not shaved. Tissue samples were taken from 5 animals per time point at T = 2 h, 72 h, 14 d, and 28 d after the placement of the LLB; for ethical reasons, the 5 control-banded animals were sampled only at T = 2 h in order to reduce the total number of animals receiving no analgesia on-study. To ensure a similar average starting body weight for each treatment group, the animals were ranked from heaviest to lightest prior to applying a random blocking factor for sampling. Each animal was only sampled at one time point.

When the calves were sampled at their corresponding time point, the LLB was removed, and a 4 mm punch biopsy (containing skin and subcutaneous tissue) was collected from the tissue that had been directly contacted by the band. The biopsies were frozen until being analyzed for the lidocaine content, as per Section 2.3 below. Furthermore, this tissue was electrocutaneously stimulated prior to applying the LLBs or control bands (i.e., T = 0) to establish a baseline response for each animal and again at T = 2 h to determine the effect of the local anesthetic; the animals’ response scores were graded as per above (Section 2.1 and Table 1). Those responsible for assessing electrostimulation response were blinded to the animals’ treatment group. Once sampling was completed for an animal, a new castration band (lacking lidocaine) was placed and oral meloxicam (1 mg/kg body weight; Solvet, Calgary, AB, Canada) was administered, which marked the conclusion of the animal’s participation in the study.

### 2.3. Assessment of Tissue Lidocaine Concentrations

Those responsible for completing the laboratory analysis of lidocaine concentration were blinded to the animals’ sampling time and treatment group. The biopsies were stored at −80 °C until processing, at which point they were thawed and placed into tubes containing 2.8 mm ceramic beads (Cat # 10158-612, VWR, Mississauga, ON, Canada) and 1 mL of mobile phase (40% *v*/*v* acetonitrile, 60% *v*/*v* Phosphate-Buffered Saline, pH 7.4). The tubes were weighed before and after the addition of the biopsies to calculate the net tissue weight. The samples were then homogenized in a FisherBrand Bead Mill 24 Homogenizer (Model No. 19-2241A) using 2 cycles (5 min per round, 6 m/s, 20 °C), with at least 1 min of cooling between rounds. The tubes were centrifuged (5 min at 12,000× *g*) to obtain the pellet cell debris before removing 800 µL of the supernatant and being passed through a 0.45 µm pore-sized nylon filter. The filtered supernatant was analyzed for the lidocaine content by HPLC using the parameters outlined in Table 2. The lidocaine concentration (mg lidocaine per mL of mobile phase) was calculated from a standard curve and converted to mg lidocaine per g (wet weight) of tissue.

### 2.4. Statistical Analysis

Descriptive statistics (mean and standard error of the mean) were generated for each study. The continuous data were assessed for normality using a Shapiro–Wilk test. For the injectable lidocaine study, the tissue lidocaine concentrations were evaluated with a repeated measures one-way ANOVA (a Dunnett’s test was used to correct for multiple comparisons), while the non-parametric data (i.e., electrostimulation response scores) were evaluated with a repeated measures Friedman test (a Dunn’s test was used to correct for multiple comparisons). Non-linear regression was used to calculate the scrotal EC_50_ and EC_95_ values by fitting the data (i.e., electrostimulation response scores versus tissue lidocaine concentration) with a variable slope sigmoidal (four-parameter logistic) dose–response curve [28]. For the in vivo delivery of lidocaine from the LLBs, a one-way ANOVA was used to evaluate the tissue lidocaine concentrations at each time point (a Dunnett’s test was used to correct for multiple comparisons), while the electrocutaneous responses at T = 2 h were compared to the corresponding baseline (i.e., T = 0) scores using a Wilcoxon matched-pairs signed-rank test. In all cases, the significance level was *p* < 0.05, while a trend was defined as a *p*-value between 0.05 and 1.0.

## 3. Results

### 3.1. Determination of the Effective Tissue Concentrations of Injectable Lidocaine

The animals displayed a minimal reaction to the lidocaine injections (i.e., minimal evasive behavior or vocalization). Lidocaine was undetected in the control animals that did not receive lidocaine. In the calves that received an injection with lidocaine, the highest concentration was at 30 min post-injection, reaching approximately 0.8 mg/g before gradually dropping to 0.04 mg/g by 240 min post-injection (Figure 2).

For all six animals, the maximum electrostimulation response score of 3 was reached at T = 0 (i.e., prior to the lidocaine injection) and dropped to 0 at T = 30 min post-injection (Figure 3). The average score increased to approximately 0.3 at 60 min post-injection and, by 90 min post-injection, no significant differences were noted from the baseline response score (Figure 3).

The tissue lidocaine concentrations from Figure 2 were plotted against the electrostimulation response scores from Figure 3 in order to calculate the EC_50_ and EC_95_ values. EC_50_ was 0.63 mg/g (95% confidence interval (CI): from 0.46 to 0.83) and EC_95_ was 1.14 (95% CI: from 0.65 to 3.34) (Figure 4).

### 3.2. In Vivo Delivery of Lidocaine into Scrotal Tissues from the LLBs

The tissue lidocaine levels in the calf scrotums approached the EC_50_ by 2 h following LLB application (the earliest sampled time point) and met or exceeded the EC_95_ between 2 and 72 h post-application (Figure 5). Moreover, the tissue lidocaine concentrations increased over time, yielding a significant linear trend (*p* = 0.0011; Figure 5), and the lidocaine concentrations were above the EC_95_ for 28 days (the final sampled time point) (Figure 5).

Relative to the baseline (i.e., pre-banding), the control-banded animals demonstrated no statistically significant decrease in electrocutaneous response scores at T = 2 h after banding (*p* = 0.25), while the LLB-treated animals tended (*p* = 0.0625) to have lower scores at T = 2 h relative to the baseline (Figure 6).

## 4. Discussion

Given the importance of animal welfare, not only for the animals, but for the sustainability of food animal industries [29], it is critical to prevent pain arising from common management procedures. The present study aimed to determine the effective tissue concentrations of the current standard of care for acute pain amelioration and evaluate the use of a novel lidocaine-impregnated band to minimize the pain associated with the castration of dairy calves.

Although lidocaine is a well-studied local anesthetic in humans [18,19,20], there is little evidence regarding its pharmacokinetics, pharmacodynamics, and effective concentrations for local anesthesia in calf scrotal tissues. Coetzee [21] and Schwartzkopf-Genswein [1] reviewed the scientific evidence on lidocaine when administered to prevent the pain associated with castration in calves; yet, very little pharmacokinetic information is provided. To the best of the authors’ knowledge, this is one of a few studies to measure any pharmacokinetic and pharmacodynamic properties—and the first to measure effective concentrations (EC_50_ and EC_95_)—of injectable lidocaine in the scrotum of calves. In this study, lidocaine was injected at multiple sites around the base of the scrotum to form a “ring block”, which has been shown to effectively mitigate the short-term pain associated with castration [25]. By scoring the animals’ response to electrocutaneous stimulation and plotting this against the tissue lidocaine concentration, the percent effective concentration was calculated by fitting the data using a variable slope sigmoidal dose–response curve, as previously described [28]. These effective concentrations are important metrics of a local anesthetic’s potency [13,14,15] and, by measuring the tissue concentration over time and comparing it to the EC_50_, the duration and time of onset of local anesthesia can be established [16,17]. Note that, although the analysts responsible for measuring tissue lidocaine levels were fully blinded to the identity of the samples, a potential limitation of this study is that the assessors responsible for scoring the animals’ reactions to electrocutaneous stimulation after lidocaine injection could not be blinded in a similar way. While a retrospective analysis of video recordings in a random order can be employed for certain behaviors, electrocutaneous stimulation requires the application of electrodes to manually restrained animals and a real-time assessment of the animal’s physical avoidance response. Thus, the best way to blind the assessors in the injection study would have been to include placebo-treated control animals at each time point, effectively doubling the number of animals on study and ensuring their electrocutaneous stimulation without analgesia, which was deemed unethical from an animal welfare standpoint. Nonetheless, this study used a binary assessment of “responded/did not respond” to the various currents applied by the various rheostat settings of the SunStim device; moreover, this binary assessment was reached by the consensus of three individuals—none of whom were involved in the injection procedure, nor aware of the contents of the syringe, nor had any vested interest (financially or otherwise) in injectable lidocaine. Therefore, we deemed the assessment of electrocutaneous response after lidocaine injection to be appropriately rigorous and robust to any observational bias.

As noted in previous studies, it was found in this paper that injected lidocaine had a short duration of action. Stewart et al. (2010) found that the injection of local anesthetic in Holstein calves prior to surgical castration led to a reduced maximal eye temperature, heart rate, and cortisol in the 25 min after conducting the procedure [30], corresponding with the peak anesthetic concentrations found in the present study. However, the duration of action in the present study was short, with tissue concentrations declining after 30 min and sensitivity to electrostimulation returning by 90 min post-injection in the current study. Others have reported similar results with Holstein calves to which local anesthetic was administered, which were found to have increased plasma cortisol at 120 min after castration when compared to those with multimodal pain relief [31]. Furthermore, in crossbred Angus calves, those that received a ring block with lidocaine had lower cortisol levels compared to those without pain control at 30 and 60 min post-castration; however, no differences were noted by 120 min post-castration [25]. Cumulatively, it is clear that the provision of injectable lidocaine has a short duration of action and, to better control pain, long-term pain relief is needed.

It is often thought that long-term pain relief can be achieved using an NSAID; however, it likely depends on the castration method used. When using a rubber band for castration, a combination of local anesthetic and NSAID can control pain over a short period after castration, yielding no formation of negative memory associated with the procedure in the 96 h after completion [32]. However, in the weeks following rubber band application, calves exhibit lower weight gains, starter intake, and lying time likely associated with a protracted pain response [12], suggesting that NSAID re-administration would be required to maintain efficacy over the duration of the elastration procedure. LLBs, on the other hand, were designed to address both acute and chronic discomfort by delivering lidocaine rapidly and for a prolonged duration. Indeed, previous work has demonstrated in vitro that LLBs release lidocaine initially rapidly (K_obs_ = 2.98 mg/h) for the first ~30 h after application, slowing to K_obs_ = 0.0292 mg/h for the next ~138 h [24]. In the present study, it was found that the use of an LLB in band castration led to lidocaine levels well above the effective concentrations of injectable lidocaine needed to reduce local sensation. Specifically, when evaluating the tissue concentration achieved using LLBs and comparing it to the EC_50_ and EC_95_ calculated for injectable lidocaine, the tissue lidocaine levels approached the EC_50_ by 2 h and met or exceeded the EC_95_ by 72 h following band application in calves. Furthermore, the concentration of lidocaine remained above these levels until at least 28 days following application (note that 28 days was the final tested time point because the animals began casting thereafter). This suggests that the LLBs could serve as a method to provide a sustained relief of pain associated with rubber band castration.

As a proof of concept, the response of scrotal tissue to electrostimulation was compared for animals treated with the control bands versus the LLBs at T = 0 (i.e., prior to band application) and T = 2 h after band placement. Importantly, those responsible for band application were blinded to the identity of the control or test bands, as were those responsible for assessing the electrocutaneous stimulation avoidance response. While the control-banded animals demonstrated no statistically significant decrease (*p* = 0.25) in the electrocutaneous response scores, the LLB-treated animals tended (*p* = 0.065) to have lower scores, suggesting that LLBs may reduce scrotal sensation when compared to the control bands lacking lidocaine. Further research with a larger sample size will be required to determine whether this effect is statistically significant. Although these results suggest the LLB could serve as a method to mitigate the long-term pain associated with castration, it is critical to have future studies conducted with larger sample sizes to provide more insights. Moreover, as the electrostimulation used in this paper to assess sensation might not produce the same type of pain as ischemia due to band castration, a large field trial comparing different pain management approaches that evaluate the behavioral (i.e., lying time and lesion licking), production (i.e., growth and feed intake), and physiological (i.e., cortisol) measures associated with pain is needed to further evaluate the efficacy of the LLBs.

## 5. Conclusions

The present study investigated the effective tissue concentrations of injectable lidocaine yielding 50% or 95% reductions in local sensation (EC_50_ and EC_95_, respectively) and identified that the use of traditional pain control—a ring block with injected lidocaine—was effective in mitigating the pain associated with castration for up to 60 min post-procedure, highlighting the importance of long-term pain control strategies. The use of a LLB led to tissue concentrations of lidocaine approaching the EC_50_ by 2 h and meeting or exceeding the EC_95_ by 72 h, with levels remaining above the EC_95_ for at least 28 days, indicating that local anesthesia is likely provided by LLBs. These data are supported by the electrostimulation response scoring of the control- versus LLB-treated animals, although the electrostimulation used in this paper to assess sensation might not produce the same type of pain as ischemia due to band castration. Further studies are therefore needed to compare the use of LLBs to injectable local anesthetics.

## Figures and Tables

**Figure 1 animals-14-00977-f001:**
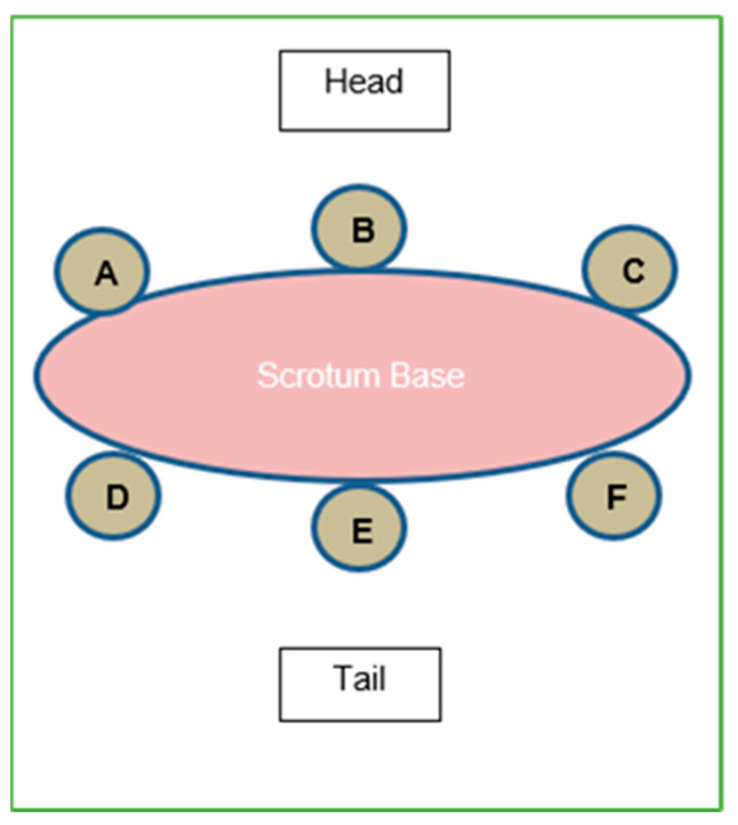
Sample injection and biopsy site layout. The locations of sites A–F were color-coded with markers and were randomized for each animal. Calves were treated with an injection of 1 mL of 2% lidocaine (without epinephrine) at each site (A–F) to create a ring block. See also reference [24].

**Figure 2 animals-14-00977-f002:**
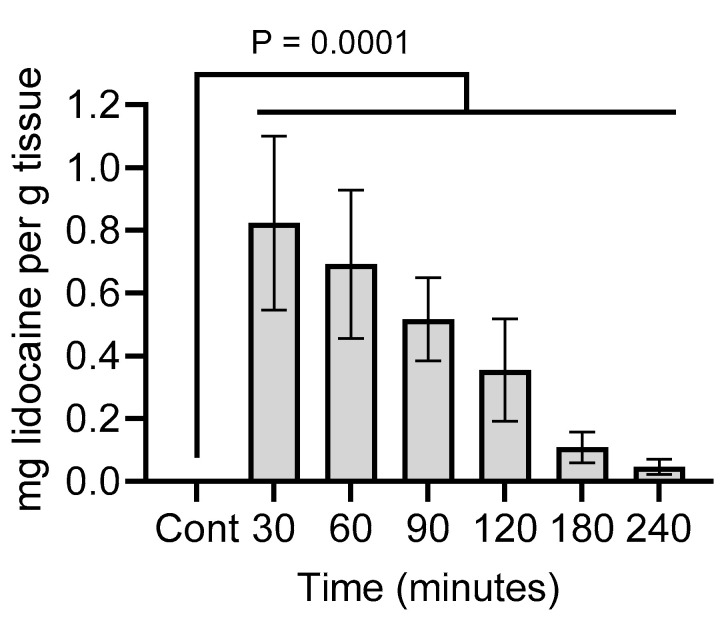
Tissue lidocaine concentrations for the calf scrotums at the indicated times after lidocaine injection and in the control animals that did not receive a lidocaine injection. Relative to the control (Cont) samples, *p*-values were determined for samples at each time point using a repeated measures one-way ANOVA (a Dunnett’s test was used to correct for multiple comparisons). Bars denote the mean ± standard error of the mean (SEM) of 6 animals.

**Figure 3 animals-14-00977-f003:**
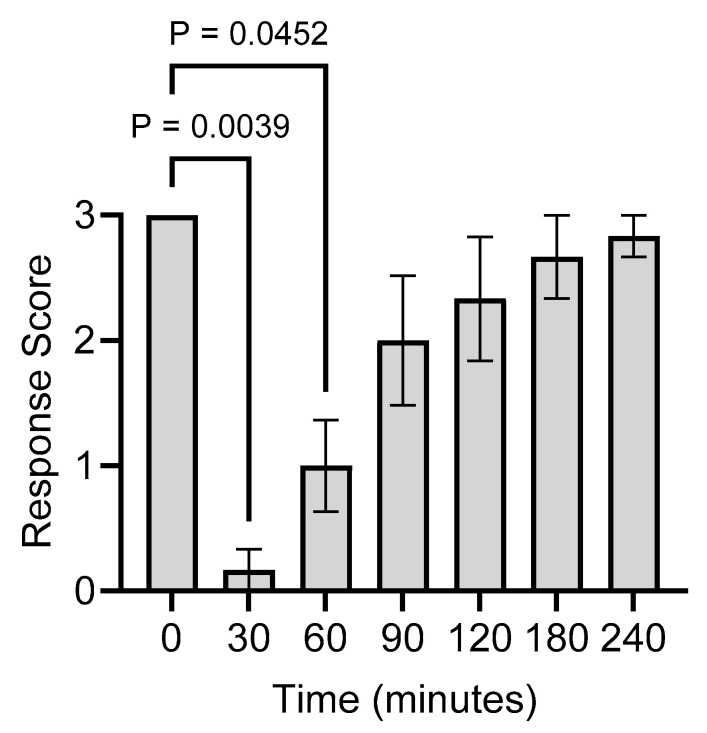
Electrocutaneous stimulation response scores, as per Table 1, for the calf scrotums at the indicated times after lidocaine injection (T = 0). Note that the response scores at each time post-injection were not normalized to the baseline scores. Relative to T = 0, *p*-values were determined at each time point using a repeated measures Friedman test (a Dunn’s test was used to correct for multiple comparisons). Bars denote the mean ± SEM of 6 animals.

**Figure 4 animals-14-00977-f004:**
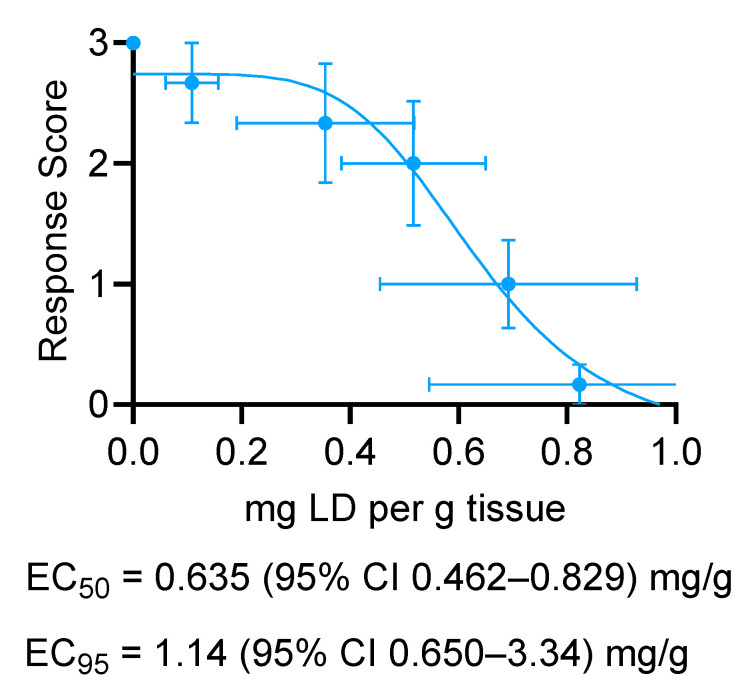
Effective concentrations of injectable lidocaine. Stimulation response scores from Figure 3 were plotted versus tissue lidocaine (LD) concentrations from Figure 2 (on the *y*- and *x*-axes, respectively), and the EC_50_ and EC_95_ values for the calf scrotal tissues were determined using non-linear regression.

**Figure 5 animals-14-00977-f005:**
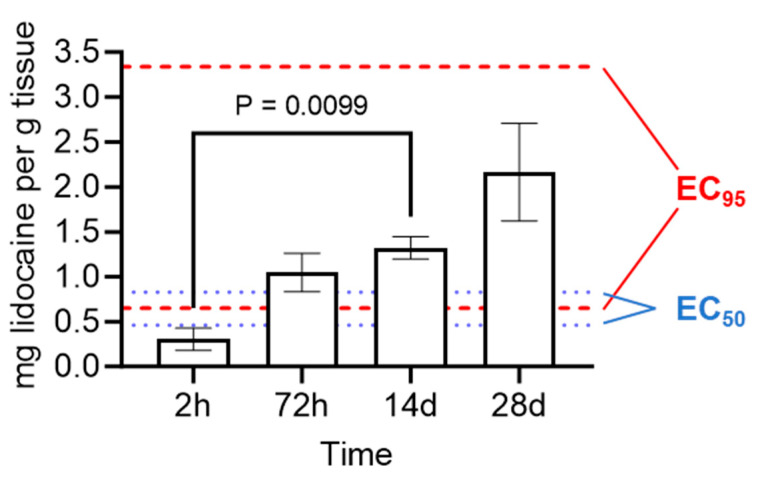
Lidocaine levels in the calf scrotal tissues after applying the LLBs. Bars denote the mean ± SEM for 5 animals; note that a different set of animals was sampled at each indicated time. *p*-values were determined using a one-way ANOVA (a Dunnett’s test was used to correct for multiple comparisons). A test for the linear trend yielded a *p*-value of 0.0011. For reference, the dashed red lines denote the upper and lower limits of the 95% CI of the EC_95_, whereas the dotted blue lines denote the upper and lower limits of the 95% CI of the EC_50_ (Figure 4). Note that no lidocaine was detected in the lidocaine-free control tissue.

**Figure 6 animals-14-00977-f006:**
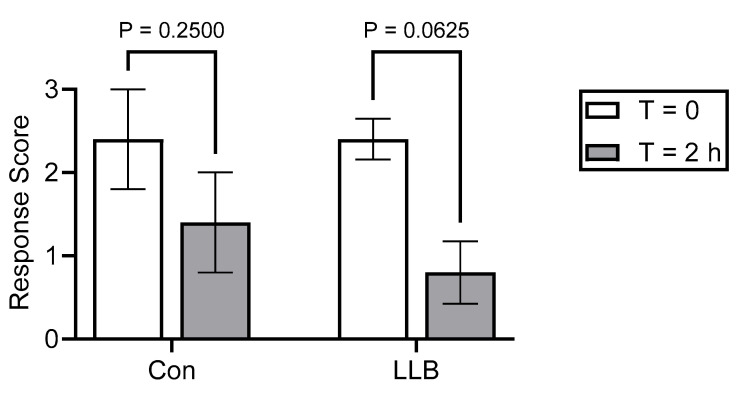
Electrocutaneous stimulation response scores for the LLB- and control-banded animals. Scores were measured at T = 0 (i.e., prior to band placement) to establish the baseline response for each individual animal, and again at T = 2 h after band placement. Bars represent the mean ± SEM of 5 animals. *p*-values were determined using a paired Wilcoxon test.

**Table 1 animals-14-00977-t001:** Electrocutaneous stimulation response scoring for tissue sensitivity.

Graded Response	Description ^a^
0	No reaction at Level 10
1	Positive reaction at Level 10
2	Positive reaction at Level 8
3	Positive reaction at Level 6 or below

^a^ Reaction was considered a positive avoidance response when the animal exhibited any (or all) of the following: side-to-side movement and tail flick; a pronounced kick or jump; bawling or vocalization; or head shaking. Note that no animals in this study reacted at stimulus levels 2 or 4.

**Table 2 animals-14-00977-t002:** HPLC specifications and running conditions (see also reference [24]).

Parameter	Details
HPLC	Agilent 1100 and 1200 Series (Agilent Technologies, Santa Clara, CA, USA)
Column	ZORBAX Extend-C18; 4.6 × 150 mm; 3.5 µm (Agilent Technologies, Santa Clara, CA, USA)
Mobile Phase	40:60—Acetonitrile:PBS, pH 7.4
Analysis Time	15 min
Flow Rate	1.0 mL/min
Injection Volume	10 µL
Column Temperature	28 °C
Detector	Agilent G1315B Diode Array Detector (DAD) (Agilent Technologies, Santa Clara, CA, USA)
Wavelength	210 nm
Bandwidth	4 nm

## Data Availability

The original contributions presented in the study are included in the article. Further inquiries can be directed to the corresponding author.

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
