# Peer review of "Assessment of the Effective Tissue Concentrations of Injectable Lidocaine and a Lidocaine-Impregnated Latex Band for Castration in Calves"

_animals, 2024, doi:10.3390/ani14060977_

Round 1

Reviewer 1 Report (Previous Reviewer 3)

Comments and Suggestions for Authors

The paper is now in a form that can be published. Looking foreward to have these bands!

Author Response

Reviewer 2 Report (Previous Reviewer 2)

Comments and Suggestions for Authors

Novel techniques and drugs for amelioration of pain in routine farm animal management procedures are needed to meet consumer demand, satisfaction and address animal welfare issues. Castration in calves is a painful procedure and pain is often overlooked leading to trauma and compromised animal welfare. Use of local anaesthetic impregnated bands for longer term application than injectable anaesthetics hold promise in ameliorating pain for a longer period of time. Research is needed for testing their efficacy for routine and extensive use.

A very well conceptualized study and flawlessly written manuscript. I congratulate the authors for this brilliant work. The statistical treatment of the data is accurate and the results have been nicely presented and well supported with graphs. Discussion is precise and well supported with current references and data. Conclusions drawn are relevant and do frankly reveal the limitations of the study and provide a path for future research on the topic.

The authors have declared their interests and I don’t think there is a conflict as industry driven research is needed keeping in mind the ethics, the authors have openly declared their interests.

Specific comments

Line 27: Please delete ‘was’

Line 119: I think ‘without’ will be a better word than ‘lacking’

Line 300: Please replace ‘both review’ with ‘reviewed’

Author Response

Reviewer 3 Report (New Reviewer)

Comments and Suggestions for Authors

Please add a couple of sentences about the housing and husbandry, including diet, of the calves during their 1 week of acclimatisation.

Line 119 and 123 – substitute ‘without’ for ‘lacking’

Author Response

This manuscript is a resubmission of an earlier submission. The following is a list of the peer review reports and author responses from that submission.

Round 1

Reviewer 1 Report

Comments and Suggestions for Authors

The Authors have submitted a paper investigating a novel method to provide local anesthesia to the scrotum of calves. The topic is relevant and the study has a number of merits, all of which would add to out knowledge within this specific perspective. However, there are some major concerns in how the manuscript has been prepared (particularly presenting methods as results, which presently makes following the manuscript very difficult). Moreover, the study design(s) need to be disclosed in a more detailed fashion and the statistical approach is unacceptable in its present form.

I would suggest, that the Authors take a close look at the submission guidelines and prepare their manuscript accordingly.

I have commented on the major issues only at this time. I sincerely hope to see this manuscript again, after a thorough revision has ben completed:

Title/general:

”Pharmacokinetics” typically refers to the fate of a drug at a systemic level, typically sampled from the central compartment. Furthermore, “pharmacodynamics” would in my opinion require the characterization of more than a single, local effect (lidocaine possesses many) . It would thus be more appropriate to refer to this particular study as e.g. ‘Target tissue concentrations and local anesthetic effects..”

Introduction:

L100 – lack of hypothesis

Material and methods:

Study design for both studies (randomized, controlled, assessor-blinded,..). Blinding is of utmost importance, given the Authors’ financial investment in the LLB technology – to be more precise, failing to describe believable and appropriate blinding would make the manuscript vulnerable to rejection in my opinion.

Moreover, the approach chosen to evaluate the animals’ response to the applied current is very confusing. First of all, the avoidance response score is an ordinal parameter and cannot be evaluated or analyzed as a parametric one. Second, a change in the response score (for the LLB part of the study) is not an appropriate way to evaluate the efficacy of the intervention – e.g. a decrease from 3 to 2 would make little difference from an animal welfare perspective. It would make much more sense simply to describe the response as positive or negative and analyze the data per this binomial. Ideally, the current tolerance would have been evaluated as the primary outcome, but it appears that this was standardized to baseline (i.e. eliciting a positive response).

L167-172 – block randomization performed for sampling times rather than treatment (4 x 5 = 20 animals?) 10 animals for control? Also, time point when the control animals were tested by electrostimulation should be detailed here, not embedded in the results. Moreover, was the electrostimulus standardized to the baseline positive response (presently this in hinted in a table legend but not mentioned elsewhere).

Statistics:

Please include power analyses and normaoity testing for the lidocaine concentrations. As mentioned above, the response data to electrostimulation needs to be reconsidered.

Results:

L229 E.g. this is a conclusion

L233-234: E.g. this should be moved to the methods section

L257-261 move to discussion

L263 “Accumulation” is the relationship between how much drug is added relative to how much is being eliminated (or systemically absorbed in this case). As both are unknown, it would be more appropriate to state that ‘concentrations increased..’ Similarly, on L265 “remained” refers to a phase of decline, which cannot be concluded from the data (it is possible that concentrations would have continued to increase after 28 days). So, again, it would preferable to rephrase to e.g. “..and the lidocaine concentrations were above the EC95..” to be more precise.

Discussion:

Not commented at this time.

Reviewer 2 Report

Comments and Suggestions for Authors

Reviewer comments for manuscript ID animals-2823174 entitled ‘Assessment of the pharmacokinetics and pharmacodynamics of injectable lidocaine and a lidocaine-impregnated latex band for castration in calves’

General comments

Novel techniques and drugs for amelioration of pain in routine farm animal management procedures are needed to meet consumer demand, satisfaction and address animal welfare issues. Castration in calves is a painful procedure and pain is often overlooked leading to trauma and compromised animal welfare. Use of local anaesthetic impregnated bands for longer term application than injectable anaesthetics hold promise in ameliorating pain for a longer period of time. Research is needed for testing their efficacy for routine and extensive use.

A very well conceptualized study and flawlessly written manuscript. I congratulate the authors for this brilliant work. The statistical treatment of the data is accurate and the results have been nicely presented and well supported with graphs. Discussion is precise and well supported with current references and data. Conclusions drawn are relevant and do frankly reveal the limitations of the study and provide a path for future research on the topic.

The authors have declared their interests and I don’t think there is a conflict as industry driven research is needed keeping in mind the ethics, the authors have openly declared their interests.

Specific comments

Line 112: Please delete ‘On June 24, 2021’

Line 127: Why lignocaine with adrenaline not used as this combination allows the prolonged effect of lignocine than the plain one? Please clarify.

Line 181: Why a new LLB was placed? This can lead to bias in the protocol vis-a vis Group 1.

Lines 159-84: How did you account for the painful effect of LLB application for few days post application? It got masked by the administration of Meloxicam or is the band an improvisation over previous ones providing immediate analgesia. Please clarify.

Reviewer 3 Report

Comments and Suggestions for Authors

Dear Authors,

very interesting important study! Nevertheless I have quiet a few things that should be ammended before the study can be accepted. Please find the specific comments here:

Generally the term Pharmacokinetics and pharmacodynamics does not seem to be justified. You simply measured local tissue concentrations and therefore the title should be local tissue concentration of lidocaine following sc injection compared to ...and concomittant local anaesthesia-.

The scientific abstract is not consise.Simply report step by step what you did in how many animals. Has to be rewritten.

Overall the description of math meth is not concise and clear.

Did you shave before applying the band.

in the sc group the 2 controll animals also were stimulated like all the others? were investigators blinded? Castration of the controll animals with knife - with no analgesia? All the biopsies were also taken in the controll animals???you really took 6 x 4mm biopsy per injection site (so a huge whole in the end?)

Description of the band group is not clear-you report to have tested 30 calves but you only report 4 time points with 5 calves each= 20 what did you do with the other ten?. So I think this part has to be rewritten carefully to make it clear (ev a figure yould help the reader to understand what you really did).

Results: you dont report at which intensity the scores were measured that you report in both groups-this seems very important!

Discussion:

you have to discuss the limitations of the study which certainly are that assessement of reaction to pain was not blinded and in the band study you had no controll animals. Also electrostimuation is certainly not causing the same pain as ischemia as present after banding testes- so this aspect should also be included in discussion (and maybe in the limitation section).

I would also find it necessary to discuss the necessity of a controll in the sc lidocaine level group.in particular form the ethical standpoint....- I think it is important to have blinded suitable controlls if we assess efficacy- but for measurement of tissue levels???

Round 2

Reviewer 1 Report

Comments and Suggestions for Authors

The Authors have provided a revision. Some modifications have been made and the manuscripts reads better and is easier to follow. However, I'm left confused as to who are these "field technicians" that were blinded to the treatment (LLB vs placebo)? Did these technicians have anything to do with the assessment of the avoidance response? If they performed the scoring, then that should be stated very clearly in the methods section. It is critical that no doubt remains that the person(s) evaluating the response to the EC stimulation were in fact blinded to the treatment. Presently the manner in which blinding is described is unacceptable and, if this issue will remain as is, I recommend rejecting this manuscript as the study design would be critically flawed.

I also remain of the opinion that delta scoring is not appropriate as the only clinically significant change would be from > 2 to < 2, all other changes would simply be statistical noise. The Authors mention Fielheller et al (2012) in their rebuttal - however they had a markedly different approach and chose a yes/no response to evaluate the efficacy of various blocks. Moreover, data cannot be treated as non-parametric on one instance and parametric in another, which is how everything related to the avoidance scores is currently presented.

I sincerely hope that the Authors will solve the blinding issue so that we can move forward with the paper.

Reviewer 3 Report

Comments and Suggestions for Authors

The manuscript has been improved but a few things in materials and methods should still be ammended AND a short phrase about limitations included. The lack  of proper blinding is a major limitation concerning the testing of efficacy.

Please find detailed comments below.

Line 138: phrase has to be deleted -should go to introduction or discussion.

Phrase line 146 “two additional animals served as negative controls; these underwent knife castration, with biopsy samples taken to provide lidocaine-free control tissue” is still not precise enough. As far as I understood comments of you in replies to reviewers you snaped of the whole scrotum during castration and then took biopsies from this tissue? So why don't you write it like this?

Line 183 I presume that no sheep was shaved- correct? Please rephrase that this is 100% clear.

Line 184/185: I would make two phrases here that it is absolutely clear. In five animals per timepoint tissue samples were taken; T., T. .-T…Samples were taken according to a random blocking factor; to ensure a similar average starting body weight for each group, the animals were weighed and ranked heaviest to lightest prior to applying the blocking factor.  Each animal was only sampled at one time point.

Line 197- maybe you add: and relevant calves were stimulated, so that it is 100% clear that only calves that were sampled were also stimulated.

Line 353-this new paragraph does not make sence. For example the reaction to stimulus could have been videotaped and retrospectively analysed in random order (including the pre local anaesthesia reaction) by somebody unaware of treatment and timing of treatment. Please delete and add as limitation comments about the lack of blinding.

Last comment to first review seems unacceptable. To castrate with knife with only a nonsteroidal anti-inflammatory agent on board should be considered highly unethical-the readers will judge about this

Round 3

Reviewer 1 Report

Comments and Suggestions for Authors

The Authors have submitted a second revision, with the complete removal of the response data from the LLB field study as the most notable modification. However, this presents a novel problem as now the reader is expected to accept tissue lidocaine concentrations as the only indicator of the drug’s efficacy. Given that the 'effective concentrations' were produced from a poorly controlled, non-blinded pilot experiment with a subjective assessment score as the only outcome variable, the suggested association between efficacy and concentration is fragile at best. Moreover, and in conflict with previous suggestions, EC50 and EC95 are still calculated as if the response score would be a parametric variable and much of the results remain erroneously presented for the same reason. And even if the data management issues would be solved, the only outcome variable would remain subjective and performed by unblinded individuals with a financial investment in the results.

Unfortunately, it has become clear that several critical flaws in how this study was designed prevents accepting this manuscript. The Authors must have been aware beforehand that their methodology would be scrutinized and yet, no substantial ad hoc steps to shield the results from inherent bias were made.

Reviewer 3 Report

Comments and Suggestions for Authors

Dear authors,

the manuscript is now clear and concise. There are still doubts about the methodology and it seems inevitable, that:

In the abstracts as well as in the conclusions it is stated that the pain elicted was by electrostimulation and not castration

otherwise it would be misleading to the reader.